# Improved Feature Importance Computation for Tree Models Based on the Banzhaf Value

**Adam Karczmarz**[1,2]     **Tomasz Michalak**[1,2]     **Anish Mukherjee**[1,2]     **Piotr Sankowski**[1,2,3]     **Piotr Wygocki**[1,3]

[1]Institute of Informatics, University of Warsaw, Poland
[2]IDEAS NCBR, Warsaw, Poland
[3]MIM Solutions, Warsaw, Poland

## Abstract

The Shapley value – a fundamental game-theoretic solution concept – has recently become one of the main tools used to explain predictions of tree ensemble models. Another well-known game-theoretic solution concept is the Banzhaf value. Although the Banzhaf value is closely related to the Shapley value, its properties w.r.t. feature attribution have not been understood equally well. This paper shows that, for tree ensemble models, the Banzhaf value offers some crucial advantages over the Shapley value while providing similar feature attributions.

In particular, we first give an optimal $O(TL + n)$ time algorithm for computing the Banzhaf value-based attribution of a tree ensemble model's output. Here, $T$ is the number of trees, $L$ is the maximum number of leaves in a tree, and $n$ is the number of features. In comparison, the state-of-the-art Shapley value-based algorithm runs in $O(TLD^2 + n)$ time, where $D$ denotes the maximum depth of a tree in the ensemble. Next, we experimentally compare the Banzhaf and Shapley values for tree ensemble models. Both methods deliver essentially the same average importance scores for the studied datasets using two different tree ensemble models (the sklearn implementation of Decision Trees or xgboost implementation of Gradient Boosting Decision Trees). However, our results indicate that, on top of being computable faster, the Banzhaf is more numerically robust than the Shapley value.

## 1 INTRODUCTION

Tree ensembles are one of the most commonly used models for solving practical problems [Friedman, 2001, Kaggle, 2017]. Tree ensembles are robust, easy to tune, and fast to train. They need small computational resources and support different types of data and missing values. Given this, tree ensembles are often the first choice model for tabular data.

One of the key research challenges regarding tree ensemble models (see Section 2 for a formal definition) and other machine learning techniques, in general, is explainability. When high-value decisions are taken, e.g., in medical diagnostic, understanding why a model made a specific prediction is often more important than the prediction's accuracy. Thus we need to develop methods to interpret the model's results in a transparent way so that humans are willing to follow model recommendations. And indeed, a large body of previous work has been devoted to explaining tree models and their predictions, e.g., [Chen and Guestrin, 2016, Breiman et al., 1984, Breiman, 2004, Brophy and Lowd, 2020, Kuralenok et al., 2019, Lundberg et al., 2020, Saabas, 2022].

Feature attribution is one of the approaches to interpreting model predictions that has been recently subject to a growing interest. In this approach, each feature's impact, or importance, on the model's output $f(x)$ is quantified using a numerical value, called the feature's *local attribution* (e.g., [Lundberg and Lee, 2017, Sundararajan et al., 2017]). Similarly, one can attempt to quantify the individual features' overall impact on the model using *global* attributions (e.g., [Covert et al., 2020, Lundberg et al., 2020]).

One of the most popular approaches to feature attribution uses methods originating from cooperative game theory that are called solution concepts or *values*. They measure the importance of each player in, or contribution to, a coalitional game. While there exist many ways in which the importance of each player can be evaluated, some solution concepts are considered more fundamental than others due to underlying axiom systems that uniquely determine them. One important game-theoretic solution concept that attracted a lot of attention in the context of explainability is *the Shapley value* (e.g., [Lundberg et al., 2020, Lundberg and Lee, 2017, Štrumbelj and Kononenko, 2014, Sundararajan et al.,

*Accepted for the 38th Conference on Uncertainty in Artificial Intelligence* (UAI 2022).

2017]). To formally introduce this concept, let us denote by $\langle g, U \rangle$, a coalition game where $g : 2^U \to \mathbb{R}$, $g(\emptyset) = 0$, is the *set function* that assigns utility to each coalition, and $U = \{1, \ldots, n\}$ is the set of players (or – in our context – features). Then, the Shapley value of the feature $i \in U$ is defined as follows [Shapley, 1953]:

$$\phi_i = \frac{1}{n} \sum_{S \subseteq U \setminus \{i\}} \binom{n-1}{|S|}^{-1} (g(S \cup \{i\}) - g(S)). \quad (1)$$

To operationalize this formula in our context, we further need to define function $g$ that extends model $f$ to all subsets of features $S \subseteq U$, i.e., $g$ allows us to drop features $U \setminus S$ of both the input $x$ and the model $f$. There are multiple alternatives of how this can be done proposed in the literature [Sundararajan and Najmi, 2020, Janzing et al., 2020]. In this paper, we focus on a popular approach by Lundberg and Lee [2017] (see Section 2 for more details). Furthermore, it should be noted that, while the Shapley value has certain attractive properties, it is evident from the above formula that, in the general case, it requires the input of exponential size (i.e., function $g$). However, in certain structured environments, when $g$ is of a convenient form or is limited in size, Shapley value can be computed in time polynomial in the number of players (features) [Deng and Papadimitriou, 1994, Greco et al., 2015, Michalak et al., 2013, Maafa et al., 2018], e.g., for tree ensemble models [Lundberg et al., 2020].

The Shapley value is not the only solution concept that has been advocated for interpreting model predictions. The *Banzhaf value* [Banzhaf, 1965] is the most well-studied alternative for Shapley value coming from the coalitional game theory and some papers indeed suggest using it for the purpose of interpreting model predictions [Datta et al., 2015, Patel et al., 2020, Sliwinski et al., 2019]. This value, also well-known and axiomatized, aggregates contributions of individual features differently:

$$\beta_i = \frac{1}{2^{n-1}} \sum_{S \subseteq U \setminus \{i\}} (g(S \cup \{i\}) - g(S)). \quad (2)$$

Mathematically, while the Shapley value is the weighted average of marginal contributions of players to coalitions, the Banzhaf value is a simple average.

Unfortunately, the difference between these two values when applied to feature attribution has not been understood well in the literature. We note that attributions based on Shapley value have been extensively studied experimentally [Lundberg et al., 2020, Lundberg and Lee, 2017, Štrumbelj and Kononenko, 2014, Sundararajan et al., 2017], whereas in the case of Banzhaf value, such studies have been done only on some basic datasets [Datta et al., 2015, Sliwinski et al., 2019, Patel et al., 2020]. Moreover, despite very high similarity of both methods, to the best of our knowledge, no comparison between them has been done on real-world data-sets, e.g., Patel et al. [2020] compares on

a single depth-3 tree, whereas Patel et al. [2021] uses both methods for vocabulary selection in different NLP tasks without directly comparing these methods. For completeness, we review other explanation methods in the supplementary material.

The primary theoretical property that distinguishes the Shapley value from the Banzhaf value, is that of so-called *Efficiency*, that the individual importances $\phi_i$ sum up to precisely $g(U)$.[1] Several authors (e.g., [Aas et al., 2021, Sundararajan and Najmi, 2020]) find a similar property desirable for attribution methods: that the attributions sum up precisely to the difference between the output of the model and the baseline/mean prediction of the model. However, this does not always seem crucial e.g., if we only want to compare impacts of individual features, and is not guaranteed by other attribution methods used in practice, e.g., LIME [Ribeiro et al., 2016]. Furthermore, it is also possible to consider the normalized Banzhaf value that satisfies Efficiency Van den Brink and Van der Laan [1998].

**Our contribution.** In this paper we partially fill the gap by providing a comprehensive analysis of the Banzhaf value, including its comparison to the Shapley value, when applied to explainability of tree ensemble models. In particular, our contributions can be summarised as follows.

We first show that, for tree ensemble models, when using the same natural set function $g$ as in [Lundberg and Lee, 2017, Lundberg et al., 2018, 2020], Banzhaf value can be computed in linear time, noticeably faster than the Shapley value. Specifically, we develop an $O(TL + n)$ time algorithm for computing the Banzhaf value-based attribution of a tree ensemble model's output. Here, $T$ is the number of trees, $L$ is the maximum number of leaves in a tree, and $n$ is the number of features. In comparison, the state-of-the-art Shapley value-based algorithm by Lundberg et al. [2018, 2020] runs in $O(TLD^2 + n)$ time, where $D$ denotes the maximum depth of a tree in the ensemble. We note that recent papers [Arenas et al., 2021, den Broeck et al., 2021] do not improve this complexity[2], but extend the method to more complex models instead.[3] We stress that our algorithm is *asymptotically optimal*, since even the description of a tree ensemble has size $\Theta(TL)$, and the output size is $\Theta(n)$.

---

[1]The Shapley and Banzhaf values satisfy similar set of axioms, except for the Banzhaf value, the Efficiency axiom is replaced with so-called *2-Efficiency* axiom.

[2]In fact, these papers only focus on proving polynomial time complexity, and neither bound nor optimize the degrees of the actual polynomials involved. Obtaining low-degree polynomial time algorithms is crucial from the practical point of view.

[3]Though, not always without loss of generality with respect to [Lundberg et al., 2018, 2020]. For example, decision trees captured by the class of boolean circuits studied in [Arenas et al., 2021] seem to forbid using a single feature for splitting multiple times on a root-leaf path of a decision tree.

On the technical level, the algorithm of Lundberg et al. [2018, 2020], reduces computing $(\phi_i)_{i=1}^n$ to finding individual *leaf contributions* to the attribution, one per each leaf/feature pair $(l, i)$ such that $i$ is used as a split feature in some ancestor of $l$. This goal is achieved using a top-down recursive algorithm whose running time is inherently $\Omega(TLD)$ (i.e., super-linear in the input size) simply because there can be $\Theta(TLD)$ such leaf/feature pairs. This bound still holds even when this approach is applied to computing a Banzhaf-value attribution. In our approach, leaf contributions are aggregated using a more efficient bottom-up dynamic programming approach, which requires only a *linear* number of auxiliary values to be computed.

In the experiments, our algorithm visibly outperforms all other algorithms, and can lead to considerable time savings when computing feature importances for decision tree-based models in practice. Moreover, we analytically prove that for trees of depths that commonly occur in practice, our algorithm for the Banzhaf value delivers numerically correct results. Similar arguments do not seem to be applicable to the most efficient algorithms computing Shapley value based attribution even for constant depth trees.

We also perform an experimental comparison of the Banzhaf and Shapley values for tree ensemble models. For four studied real-world datasets and using two different approaches to training tree models, we verify experimentally that both methods deliver essentially the same average feature importance scores (called *global impacts* in [Lundberg et al., 2020]) and very close attributions of individual predictions despite the differences in the sets of axioms the Banzhaf and the Shapley values satisfy. However, the Banzhaf value is more numerically robust than the Shapley value, and only very small errors are observed in the computations. Overall, our analysis indicates that for tree models, the Banzhaf value has two important advantages over the Shapley value. While both methods deliver comparable attributions, the Banzhaf value works faster and is less prone to numerical errors.

## 2  PRELIMINARIES

Let $U := \{1, \ldots, n\}$ be a set of *features*. Let $x$ be the input to the model to be explained. For $i \in U$, we write $x_i$ to refer to the *value* of the $i$-th feature in $x$. More generally, for any subset $S \subseteq U$ we write $x_S$ when referring to the vector $(x_i)_{i \in S}$. We sometimes talk about random feature vectors, or consider the values of individual features to be random variables. We then write $X$ or $X_i$ respectively. We write $X_S$ to denote the vector of random variables $(X_i)_{i \in S}$. Let $\bar{S}$ denote the complement $U \setminus S$.

**Tree models.**  Let $f : \mathbb{R}^U \to \mathbb{R}$ be the output function of the model to be explained. We focus on tree ensemble models $(\mathcal{T})_{i=1}^T$ where the output $f(x)$ of the model is simply the average output $f_{\mathcal{T}_i}(x)$ of its $T$ individual trees. Following

Lundberg et al. [2020], we assume the individual trees to have the number of leaves bounded by $L$ and depth bounded by $D$.[4] Let us denote by $\rho_i$ the root of the tree $\mathcal{T}_i$.

When talking about an input decision tree $\mathcal{T}$, we adopt the notation of [Lundberg et al., 2020]. $\mathcal{T}$ is a binary tree based on single-variable splits: each non-leaf node $v \in \mathcal{T}$ is assigned a *feature* $d_v$, and a *threshold* $t_v$, whereas each leaf $l$ is assigned a real *value* $f(l)$. Let $a_v, b_v$ denote the left and right children of a non-leaf node $v \in \mathcal{T}$. The output $f_{\mathcal{T}}(x)$ of the tree $\mathcal{T}$ is computed by following a root-leaf path in $\mathcal{T}$: at a non-leaf node $v \in \mathcal{T}$, we descend to $a_v$ if $x_{d_v} < t_v$, or to $b_v$ otherwise. When a leaf is reached, its value is returned. Denote by $\mathcal{L}(\mathcal{T})$ the set of leaves of $\mathcal{T}$. Denote by $\mathcal{T}[v]$ the subtree of $\mathcal{T}$ rooted at $v$.

**Set functions.**  We write $f(x_S, X_{\bar{S}})$ when referring to a random variable being the value of $f$ if the values for features in $S$ are fixed to the respective values of $x$, and the values $X_{\bar{S}}$ are random variables. Let $X_U$ be distributed[5] as in the training set of the model $f$. Recall that a *set function* $g : 2^U \to \mathbb{R}$ with $g(\emptyset) = 0$, has to be fixed to talk about the Shapley or Banzhaf value-based attributions $(\phi_i)_{i \in U}$ and $(\beta_i)_{i \in U}$ as defined in Equations (1) and (2), resp., Lundberg et al. [2020] and Janzing et al. [2020] suggest using the following idealized[6] set function $g^*$ for feature attribution:

$$g^*(S) := \mathbb{E}[f(x_S, X_{\bar{S}})] - \mathbb{E}[f(X_U)]. \qquad (3)$$

Note the term $\mathbb{E}[f(X_U)]$ in (3) serves the purpose of having $g(\emptyset) = 0$ and cancels out when computing the Shapley value from Equation (1). Thus, for simplicity in the following we can redefine $g^*(S) := \mathbb{E}[f(x_S, X_{\bar{S}})]$.

Using the idealized set function $g^*$ would be computationally too costly. Consequently, Lundberg et al. [2020] in their TREESHAP_PATH[7] algorithm considers the set function $g$ whose purpose is to "approximate" $g^*$. Namely, $g(S) \approx g^*(S)$ is computed as shown in Algorithm 1. This method dates back to the classical work of Friedman [2001] and is also implemented as a way to compute partial dependence plots in the scikit-learn package [Pedregosa et al., 2011]. Its one advantage is that it does not require access to the training data, but merely to the "coverages" $r_v$ of all the subtrees $\mathcal{T}[v]$ (for all trees $\mathcal{T}$ in the ensemble), i.e., the

---

[4]This is merely for clarity of the obtained time bounds. See discussion after Theorem 1.

[5]In fact, here we can use any other distribution, possibly over some different validation data, such that the expectations $\mathbb{E}[f(x_S, X_{\bar{S}})]$ can be estimated using Algorithm 1. This allows us to produce attributions that are contrastive to other baselines than the mean prediction over the training data.

[6]It might seem that using marginal expectation instead of conditional expectation here leads to inclusion of unrealistic data when features are highly dependent. However, Janzing et al. [2020] gave some compelling reasons why this is still a reasonable choice.

[7]We will sometimes use an abbreviated name TREESHAP.

**Algorithm 1** Estimating $\mathbb{E}[f(x_S, X_{\bar{S}})]$

1: **function** DESC$(S, v)$
2:    **if** $v$ is a leaf **then**
3:       **return** $f(v)$
4:    **if** $d_v \in S$ **then**
5:       **if** $x_{d_v} < t_v$ **then**
6:          **return** DESC$(S, a_v)$
7:       **else**
8:          **return** DESC$(S, b_v)$
9:    **else**
10:       **return** $\frac{r_{a_v}}{r_v} \cdot$ DESC$(S, a_v) + \frac{r_{b_v}}{r_v} \cdot$ DESC$(S, b_v)$
11: **function** $g(S)$
12:    **return** $\frac{1}{T} \cdot \sum_{i=1}^{T}$ Desc$(S, \rho_i)$

numbers of training set points that fall into $\mathcal{T}[v]$. It can be proved that this method approximates $\mathbb{E}[f(x_S, X_{\bar{S}})]$ well if the individual feature random variables $X_i$ are independent. With such a set function $g$, Lundberg et al. [2018, 2020] show how to compute the Shapley value attributions $(\phi_i)_{i \in U}$ exactly in $O(TLD^2 + n)$ time.

In the remaining part of the paper, we denote by $g(S)$ the output of Algorithm 1 for the subset $S \subseteq U$, i.e., we consider the same approximation of $g^*(S)$ as in the TREESHAP_PATH algorithm of Lundberg et al. [2020].

## 3 THE BANZHAF VALUE ALGORITHM

In this section, we introduce an optimal $O(TL + n)$ time algorithm, called BANZHAF, for computing attributions based on the Banzhaf value. For clarity, let us assume first that there is just a single tree $\mathcal{T}$ in the model, i.e., $T = 1$. This is without loss of generality, since the prediction of an ensemble model is simply the average of the predictions produced by individual trees. We describe the algorithm for arbitrary $T$ later on. Due to space constraints, the proofs of technical lemmas can be found in the supplementary material.

Let $\rho$ denote the root of $\mathcal{T}$, and $p_v$ the parent of node $v \in \mathcal{T}$, $v \neq \rho$. Furthermore, let $F_v$ be the set of features assigned to the ancestors of $v$, i.e., $F_\rho = \emptyset$, and $F_v = F_{p_v} \cup \{d_{p_v}\}$ for $v \neq \rho$. The value $P[v] = r_v / r_\rho$ can be thought as the probability that the model returns a value from $\mathcal{T}[v]$.

Algorithm 1 computes the estimate $\mathbb{E}[f(x_S, X_{\bar{S}})]$. Observe that the output of this algorithm for $S = \emptyset$ is precisely equal to $\sum_{l \in \mathcal{L}(\mathcal{T})} P[l] \cdot f(l)$. More generally, denote by $P[v, S]$ the weight from the ancestor recursive calls assigned to the subtree rooted at $v$ when running Algorithm 1 with an arbitrary $S \subseteq U$. Formally, $P[\rho, S] = 1$, and for any $v \neq \rho$,

$$P[v, S] = \begin{cases} P[p_v, S] \cdot \frac{r_v}{r_{p_v}} & \text{if } d_{p_v} \notin S, \\ P[p_v, S] \cdot [x_{d_{p_v}} < t_{p_v}] & \text{if } d_{p_v} \in S, v = a_{p_v}, \\ P[p_v, S] \cdot [x_{d_{p_v}} \geq t_{p_v}] & \text{if } d_{p_v} \in S, v = b_{p_v}. \end{cases}$$

Then, the algorithm outputs

$$\sum_{l \in \mathcal{L}(\mathcal{T})} P[l, S] \cdot f(l) = g(S) \approx g^*(S). \qquad (4)$$

In our approach, each of the desired attributions $\beta_i$ is obtained by summing the contributions of each individual leaf $l \in \mathcal{L}(\mathcal{T})$ to the sum (2) with $g$ defined as in (4). More precisely:

$$\beta_i = \sum_{l \in \mathcal{L}(\mathcal{T})} \left( \frac{f(l)}{2^{n-1}} \sum_{S \subseteq U \setminus \{i\}} (P[l, S \cup \{i\}] - P[l, S]) \right).$$

We now introduce the following crucial intermediate values that will enable us to evaluate the above formula efficiently. For any $v \in \mathcal{T}$, and subset $G \subseteq U$, let

$$\beta(v, G) := \frac{1}{2^{|G|}} \sum_{S \subseteq G} P[v, S]. \qquad (5)$$

**Lemma 1.** *For any $i \in U$, we have:*

$$\beta_i = \sum_{\substack{l \in \mathcal{L}(\mathcal{T}) \\ i \in F_l}} 2f(l) \cdot (\beta(l, F_l) - \beta(l, F_l \setminus \{i\})).$$

Lemma 1 reduces computing the Banzhaf value to computing $O(L)$ values of the form $\beta(l, F_l)$, and $O(L \cdot D)$ values of the form $\beta(l, F_l \setminus \{i\})$, for all $(l, i)$ such that $l \in \mathcal{L}(\mathcal{T})$ and $i \in F_l$. The $O(L \cdot D)$ bound follows since each leaf has no more than $D$ ancestors, which implies $|F_l| \leq D$.

In the following part of the section, we first give a recursive formula for computing the values $\beta(v, G)$ efficiently using dynamic programming. Next, we show a simpler $O(LD)$ time algorithm computing all the values $\beta(\cdot, \cdot)$ required by Lemma 1. As a final step, we show how to improve the worst-case running time of the algorithm to optimal $O(L)$.

**Recurrence.** To proceed, we will need the auxiliary values $\Delta_{v,y}$ for $v \in \mathcal{T}$ and $y \in U$, defined inductively as follows:

$$\Delta_{v,y} = \begin{cases} 1 & \text{if } v = \rho, \\ \Delta_{p_v,y} & \text{if } d_{p_v} \neq y \text{ and } v \neq \rho, \\ \Delta_{p_v,y} \cdot [x_y < t_{p_v}] \cdot \frac{r_v}{r_{p_v}} & \begin{array}{l}\text{if } d_{p_v} = y \text{ and} \\ a_{p_v} = v \neq \rho,\end{array} \\ \Delta_{p_v,y} \cdot [x_y \geq t_{p_v}] \cdot \frac{r_v}{r_{p_v}} & \begin{array}{l}\text{if } d_{p_v} = y \text{ and} \\ b_{p_v} = v \neq \rho.\end{array} \end{cases}$$

The above auxiliary values can be in turn used to recursively compute the values $P[\cdot, \cdot]$.

**Lemma 2.** *Let $v \in \mathcal{T}$ and $Q \subseteq U$ and $y \in U \setminus Q$. Then:*

$$P[v, Q \cup \{y\}] = P[v, Q] \cdot \Delta_{v,y}.$$

**Algorithm 2** Computing $\beta[l] = \beta(l, F_l)$ for all $l \in \mathcal{L}(\mathcal{T})$.

---

 1: **procedure** TRAVERSE(v)
 2:     **if** $d_{p_v} \in F$ **then**
 3:         present := **true**        ▷ record that $d_{p_v}$ in $F_{p_v}$
 4:         $b := \frac{2}{1+\delta[d_{p_v}]} \cdot \beta[p_v]$   ▷ $b = \beta(p_v, F_{p_v} \setminus d_{p_v})$
 5:     **else**
 6:         present := **false**
 7:         $F := F \cup \{d_{p_v}\}$        ▷ ensure $F = F_v$
 8:         $b := \beta[p_v]$        ▷ $b = \beta(p_v, F_{p_v} \setminus d_{p_v})$
 9:     $\delta_{\text{old}} := \delta[d_{p_v}]$
10:     **if** $v = a_{p_v}$ **then**
11:         $\delta[d_{p_v}] := \delta[d_{p_v}] \cdot [x_y < t_{p_v}] \cdot \frac{r_v}{r_{p_v}}$
12:     **else**
13:         $\delta[d_{p_v}] := \delta[d_{p_v}] \cdot [x_y \geq t_{p_v}] \cdot \frac{r_v}{r_{p_v}}$
14:     $\delta^*[v] := \delta[d_{p_v}]$     ▷ store $\Delta_{v,d_{p_v}}$ for future use
15:     $b := b \cdot r_v / r_{p_v}$        ▷ $b = \beta(p_v, F_v)$
16:     $\beta[v] := b \cdot \frac{1}{2}(1 + \delta[d_{p_v}])$     ▷ Lemma 3
17:     **if** $v \notin \mathcal{L}(\mathcal{T})$ **then**
18:         TRAVERSE($a_v$)
19:         TRAVERSE($b_v$)
20:     **if** present = **false then**   ▷ revert changes to $F, \delta$
21:         $F := F \setminus d_{p_v}$
22:     $\delta[d_{p_v}] := \delta_{\text{old}}$

---

Lemma 2 applied to (5) allows computing the values $\beta(v, G)$ recursively, as stated in the below lemmas.

**Lemma 3.** *Let $v \in \mathcal{T}$ and $G \subseteq U$. Let $y \in U \setminus G$. Then:*

$$\beta(v, G \cup \{y\}) = \frac{1}{2}(1 + \Delta_{v,y})\,\beta(v, G).$$

**Lemma 4.** *Let $v \in \mathcal{T}$, $v \neq \rho$. Then, for any $Q \subseteq U \setminus \{d_{p_v}\}$,*

$$\beta(v, Q) = \beta(p_v, Q) \cdot \frac{r_v}{r_{p_v}}.$$

## 3.1 BASIC ALGORITHM

Equipped with Lemmas 3 and 4, one can easily move between "nearby" values $\beta(G, v)$. Namely, for any $i \in U$, given $\beta(v, G)$ and $\Delta_{v,i}$, each of the values $\beta(a_v, G)$, $\beta(b_v, G)$, $\beta(v, G \cup \{i\})$ can be computed in $O(1)$ time.

Moreover, the values $\beta(p_v, G)$, $\beta(v, G \setminus \{i\})$ can also be obtained in $O(1)$ time by applying the respective "inverse" forms of these lemmas. We now stress that being able to compute $\beta(v, G \setminus \{i\})$ out of a value of the form $\beta(v, G)$, i.e., removing elements from the feature set $G$, is crucial for two reasons. First, recall that we need to obtain values of the form $\beta(l, F_l \setminus \{i\})$ for all leaves $l$ and all $i \in F_l$. For all such $i$, this value can be obtained using a single inverse application of Lemma 3. Moreover, applying Lemma 4 to

obtain $\beta(v, F_v)$ out of the parent value $\beta(p_v, F_{p_v})$ requires $d_{p_v} \notin F_{p_v}$. This may be violated if $F_v = F_{p_v}$, i.e., $d_{p_v}$ is a feature in some other ancestors of $v$ in the tree (which does happen in practical models). In such a case, the inverse Lemma 3 can be used to first compute $\beta(p_v, F_v \setminus \{d_{p_v}\})$, then we apply Lemma 4 to obtain $\beta(v, F_v \setminus \{d_{p_v}\})$, and finally we again use Lemma 3 to get $\beta(v, F_v)$.

The basic algorithm (which is similar in its essence to TREESHAP_PATH), computes all the values $\beta(v, F_v)$ for $v \in \mathcal{T}$ – as explained above – using a simple recursive tree traversal in $O(L)$ time. In particular, this also gives all the values $\beta(l, F_l)$ that we need when invoking Lemma 1. Afterwards, for each leaf $l \in \mathcal{T}$, the remaining (again, required by the formula in Lemma 1) $|F_l|$ values of the form $\beta(l, F_l \setminus \{i\})$ for $i \in F_l$ can be computed in $O(1)$ extra time each using Lemma 3. As a result, through all pairs $(l, i)$, this takes $O\left(\sum_{l \in \mathcal{L}(\mathcal{T})} |F_l|\right) = O(LD)$ time.

The above analysis silently assumed that all the needed auxiliary values $\Delta_{v,y}$ can be accessed in $O(1)$ time. We now justify this assumption. During the tree traversal we store a global array $\delta$ indexed with the features $U$. We maintain an invariant that $\delta[y]$ equals $\Delta_{p_v,y}$ when the processing of a vertex $v$ starts and also when it finishes. By (3), to guarantee the invariant is satisfied upon the recursive traversals of the subtrees rooted at $a_v$ or $b_v$, we may possibly need to update only the value $\delta[d_v]$ according to (3), because $\Delta_{v,y} \neq \Delta_{a_v,y}$ or $\Delta_{v,y} \neq \Delta_{b_v,y}$ may only happen when $y = d_v$. When a recursive traversal returns, we revert that change to $\delta[d_v]$.

The pseudocode of a recursive procedure TRAVERSE computing all the values $\beta(l, F_l)$, which we also require in our optimal algorithm, is given as Algorithm 2. In this procedure, each of the computed values $\beta(v, F_v)$ is recorded in a global array as $\beta[v]$. The auxiliary global variable $F$ stores the set $F_v$ when node $v$ is processed; $F$ can be implemented using a bitmap of size $n$.

## 3.2 THE OPTIMAL ALGORITHM

The high-level idea behind our improved algorithm is to avoid computing all the leaf contributions to the individual components $\beta_i$ of the Banzhaf value separately. Instead, for every node $v \in \mathcal{T}$, $v \neq \rho$, such that $d_{p_v} = i$, we compute the total contribution to $\beta_i$ of *all* the leaves $\mathcal{L}_v \subseteq \mathcal{T}[v]$, defined to be the subset of leaves for which $v$ constitutes the *nearest* weak ancestor (i.e., a node is considered its own ancestor) with $d_{p_v} = i$, at once.

Note that for a given $i \in U$, the sets $\mathcal{L}_v$ for $v \in \mathcal{T}$ satisfying $d_{p_v} = i$, are pairwise disjoint, and in fact form a partition of the set $\{l \in \mathcal{L}(\mathcal{T}) : i \in F_l\}$ through which summation in Lemma 1 is performed. Additionally, observe that the values $\Delta_{l,d_{p_v}}$ are equal to $\Delta_{v,d_{p_v}}$ for all leaves $l$ in $\mathcal{L}_v$.

**Algorithm 3** Computing the values $B(v)$ for all $v \in \mathcal{T}$.

1: **procedure** FAST(v)
2:     $H[d_{p_v}]$.PUSH$(v)$
3:     **if** $v \in \mathcal{L}(\mathcal{T})$ **then**
4:         $S[v] := f(v) \cdot \beta[v]$
5:     **else**
6:         FAST$(a_v)$
7:         FAST$(b_v)$
8:         $S[v] := S[a_v] + S[b_v]$
9:     $z := 0$               $\triangleright$ $z$ stores the sum $\sum_{w \in Q_v} S(w)$
10:    **while** $H[d_{p_v}]$.TOP$() \neq v$ **do**
11:        $z := z + S[H[d_{p_v}].\text{TOP}()]$
12:        $H[d_{p_v}]$.POP$()$
13:    $B[v] := S[v] - z$
14:    **if** $|H[d_{p_v}]| = 1$ **then**      $\triangleright$ empty $H[d_{p_v}]$ if $g_v = \bot$
15:        $H[d_{p_v}]$.POP$()$

Consider the following values for all $v \in \mathcal{T}, v \neq \rho$:

$$B(v) = \sum_{l \in \mathcal{L}_v} f(l) \cdot \beta(l, F_l).$$

The below lemma shows that computing the Banzhaf value $\beta$ can be reduced, in linear time, to computing all the values $B(v)$, $v \in \mathcal{T} \setminus \{\rho\}$: indeed, each $B(v)$ appears in the sum below for precisely one $i \in U$.

**Lemma 5.** *For any $i \in U$, we have:*

$$\beta_i = \sum_{\substack{v \in \mathcal{T} \setminus \{\rho\} \\ d_{p_v} = i}} \frac{2(\Delta_{v,i} - 1)}{1 + \Delta_{v,i}} \cdot B(v).$$

We have previously showed that the values $\beta(l, F_l)$ can be computed in linear time. We now describe a recursive procedure FAST$(u)$, where $u \neq \rho$, computing $B(v)$ for all $v \in \mathcal{T}[u]$ in a bottom-up manner. Let

$$S(v) = \sum_{v \in \mathcal{L}(\mathcal{T}[v])} f(l) \cdot \beta(l, F_l),$$

that is, $S(v)$ sums the values $f(l) \cdot \beta(l, F_l)$ in $\mathcal{T}[v]$. Clearly, for each $l \in \mathcal{L}(\mathcal{T})$, we have $S(l) = f(l) \cdot \beta(l, F_l)$, and for a non-leaf $v \in \mathcal{T}$, $S(v) = S(a_v) + S(b_v)$ holds. As a result, all the values $S(v)$ can be computed in linear time using a bottom-up computation over the tree.

Given the sums $S(v)$, we proceed as follows. For $v \in \mathcal{T}$, let $Q_v$ be the set of non-leaf nodes $w \in \mathcal{T}[v]$ with $d_{p_w} = d_{p_v}$ and $v$ is the nearest ancestor of $w$ with $d_{p_w} = d_{p_v}$. We have: $\mathcal{L}_v = \mathcal{L}(\mathcal{T}[v]) \setminus \left( \bigcup_{w \in Q_v} \mathcal{L}(\mathcal{T}[w]) \right)$, and thus

$$B(v) = S(v) - \sum_{w \in Q_v} S(w).$$

**Algorithm 4** Computing the attributions $(\beta_i)_{i=1}^n$ of the tree ensemble model's $(\mathcal{T}_j)_{j=1}^T$ prediction $f(x)$.

1: **function** BANZHAFATTRIBUTION$(n, (\mathcal{T}_j)_{j=1}^T)$
2:     **for** $i \in U$ **do**               $\triangleright$ initialize global data
3:         $\beta_i := \beta[i] := 0$      $\triangleright$ $(\beta_i)_{i=1}^n$ stores the result
4:         $\delta[i] := 1$
5:         $H[i] = $ empty stack
6:     $F := \emptyset$
7:     **for** $j = 1, \dots, T$ **do**
8:         $\rho := $ the root node of $\mathcal{T}_j$
9:         **for** $v \in \{a_\rho, b_\rho\}$ **do**
10:            TRAVERSE$(v)$
11:            FAST$(v)$
12:        **for** $v \in \mathcal{T}_j \setminus \{\rho\}$ **do**
13:            $\beta_{d_v} := \beta_{d_v} + \frac{2(\delta^*[v]-1)}{1+\delta^*[v]} \cdot B[v]$ $\triangleright$ Lemma 5
14:    **return** $(\beta_i/T)_{i=1}^n$    $\triangleright$ average through the $T$ trees

Observe that the total size of sets $Q_v$ (over all $v \in \mathcal{T}$) is $O(L)$, so if we are allowed to iterate through $Q_v$ whenever we wish to compute $B(v)$, the computation of $B(v)$ takes $O(L)$ time as well. We now explain how to accomplish this. Let $g_w$ denote the nearest ancestor of $w \in \mathcal{T}$ with $d_{p_w} = d_{p_{g_w}}$. One way to enable iterating through $Q_v$ when $v$ is processed bottom-up, is to maintain, for each feature $j \in U$, a global stack $H[j]$ containing all the nodes $w$ such that $d_{p_w} = j$ and that the computation for $w$ (i.e., the call FAST$(w)$) has already been started or completed, but the computation for $g_w$ has not yet completed. The stack elements are sorted using the pre-order of the nodes of $v$, so that the node $w$ with the highest pre-order is at the top of $H[d_{p_w}]$. The stack can be updated in $O(1)$ time whenever a recursive call starts. Observe that $v \in H[d_{p_v}]$ when FAST$(v)$ has started but has not yet finished. Now, given $H[d_{p_v}]$, it is enough to note that $Q_v$ equals precisely the set of elements of $H[d_{p_v}]$ closer to the top of the stack than $v$. Thus, one can indeed iterate through $Q_v$ in $O(|Q_v|)$ time as desired. Moreover, $Q_v$ constitutes precisely the set of elements that have to be popped from $H[d_{p_v}]$ when FAST$(v)$ returns. The asymptotic cost of popping stack elements can charged to the corresponding pushes and thus can be neglected.

A pseudocode of the procedure FAST computing all the values $B(v)$ given the values $\beta(l, F_l)$ is given in Algorithm 3. In Algorithm 4 we give a pseudocode of the full algorithm computing the Banzhaf value-based attributions for a tree ensemble model $(\mathcal{T}_j)_{j=1}^T$. Since the value of such a model is defined to be the average prediction over all the individual tree predictions, the final attribution is simply the average of the individual attributions. We have thus proved:

**Theorem 1.** *Let $n = |U|$. The Banzhaf value-based attribution $(\beta)_{i \in U}$ of a prediction of a tree ensemble model consisting of $T$ trees with at most $L$ leaves each, can be computed in optimal $O(TL + n)$ time.*

We remark that if the ensemble contains $T$ trees of very different sizes, the time can be more precisely bounded by $O\left(\sum_{i=1}^{T} |\mathcal{T}_i| + n\right)$, i.e., remains optimal in the input size.

Finally, it is worth noting that the above approach to speeding-up the basic algorithm can be also successfully applied to reduce the time complexity of the `TREESHAP_PATH` attribution algorithm of Lundberg et al. [2020] from $O(TLD^2+n)$ to $O(TLD+n)$. This is desribed in detail in the supplementary material,

## 4  EXPERIMENTAL ANALYSIS

The goals of our experiments are threefold:

- *Time performance* — first, we test the performance of the `BANZHAF` algorithm proposed in the previous section and compare it to the performance of the `TREESHAP_PATH` algorithm by Lundberg et al. [2020]—the state-of-the-art algorithm for the Shapley value attributions for tree models.

- *Qualitative differences* — next, we investigate whether the Banzhaf value returns qualitatively different results than the Shapley value for tree models.

- *Numerical accuracy* — finally, we compare numerical accuracy of both algorithms.

### 4.1  EXPERIMENTAL SETUP AND DATASETS

In our experiment we use both the sklearn implementation of Decision Trees (DT) or xgboost implementation of Gradient Boosting Decision Trees (GBDT). These are some of the most popular algorithms for generating decision trees and are quite often used for large depths of trees. Using large-depth trees is particularly beneficial for datasets with many features and complex relationship between them (see e.g., [Bordag et al., 2021, Pham et al., 2019] for a usage of trees of depth 100). Let us emphasize that large depth of a tree, e.g. depth 100, does not mean the size of the tree is $2^{100}$, because trees might be (and usually are) unbalanced. To simplify the experiments and reduce the their running times, we trained the DT algorithm to generate only one tree. We use four "real-world" datasets (see Table 1 for key details):

1. `BOSTON` (abbr. `BS`). [BS]. This small prediction dataset contains information concerning housing in the area of Boston Massachusetts. The task is to predict the price of the house.

2. `NHANES` (`NH`). The same dataset that was used in previous work on tree model interpretability [Lundberg et al., 2020] which our work most closely relates to. The parameters used for training were the same as in [Lundberg et al., 2020].

| name | rows | cols | tree depth | iter. | max depth | learning rate |
|---|---|---|---|---|---|---|
| BOSTON | 506 | 13 | 10 | 100 | 6 | 0.01 |
| NHANES | 8023 | 79 | 40 | 250 | 4 | 0.2 |
| VEH.INS. | 304887 | 14 | 60 | 250 | 4 | 0.2 |
| FLIGHTS | 1543718 | 647 | 100 | 250 | 10 | 0.2 |

Table 1: The sizes of datasets and parametrisation of the experiments. The "tree depth" column reports tree_depth of the decision tree (DT) with all the other parameters set to default values. The "iterations", "max depth" and "learning rate" columns are the parameters used for training xgboost.

| | BANZHAF | TREESHAP | | BANZHAF | TREESHAP |
|---|---|---|---|---|---|
| BS_GB | 0.48 s | 0.70 s | BS_DT | 0.41 s | 0.41 s |
| VI_GB | 23.63 s | 35.32 s | NH_DT | 3.57 s | 42.87 s |
| NH_GB | 50.20 s | 1 m 28 s | VI_DT | 4 m 55 s | 30 m 55 s |
| FL_GB | 13 m 18 s | 48 m 8 s | FL_DT | 14 m 28 s | 5 h 9 m |

Table 2: Running times of the two attribution algorithms on the entire dataset. We observe that `BANZHAF` is substantially faster than `TREESHAP_PATH` on each instance.

3. `VEHICLE_INSURANCE` (`VI`). [VI]. A medium size dataset for predicting who might be interested in vehicle insurance based on health insurance data.

4. `FLIGHTS` (`FL`). [FL]. A large dataset for predicting the flights' delays. A large number of columns was caused by one-hot encoding 'UniqueCarrier', 'Origin', 'Dest', 'CancellationCode' in a standard way, i.e., for each possible value $v$ of a given column $c$ we created additional categorical column $c\_v$ ($v \in \{0, 1\}$) indicating that the value of $c$ equals $v$ iff the value of $c\_v$ equals 1.

We will refer to the above datasets by adding "DT" and "GB" suffixes (for DT and GBDT algorithms, resp.) to the ordinal name of the prediction dataset. Note that the parameters were not extensively tuned since our main goal here centers around interpreting models and not optimizing them.

All our experiments were performed using Intel(R) Xeon(R) CPU E5-2630 v4 @ 2.20GHz with 512 Gb of RAM using only one thread for computation. The operating system was Ubuntu 18.04.2 LTS. Our linear-time `BANZHAF` algorithm was implemented in C++, whereas for `TREESHAP_PATH`, we used to original C language implementation from the SHAP package [SHAP]. The binaries were compiled using clang version 6.0.0-1ubuntu2 with -O3 optimization.

### 4.2  COMPARISON OF RUNNING TIMES

In this section, we compare the running times of the algorithms. For each of the instances, the task was to compute the attributions of *all* individual data points. In Table 2 we show

the running times for different examples. We conclude that `BANZHAF` is consistently faster than `TREESHAP_PATH`, and using it can lead to considerable time savings for larger data-sets. As anticipated by the theoretical worst-case time complexity analysis, the observed speed-up increases with the depth of trees in the model.

## 4.3 COMPARISON OF FEATURE SCORES

We test whether the Banzhaf value assigns qualitatively different importance to features than the Shapley value. The comparison is performed from two viewpoints.

**Global importance.** First, we compare the global importances of individual features for the model. To this end, we apply the same measure of *global impact* of a feature as in [Lundberg et al., 2020]. Let $\mathcal{D}$ be some dataset. Suppose for each $i \in U$ we have some feature attribution function $\gamma_i : \mathcal{D} \to \mathbb{R}$. Let us consider the global impact of the feature over dataset $\mathcal{D}$ measured as $\Gamma_i = \sum_{x \in \mathcal{D}} |\gamma_i(x)|$. For example, we can set $\gamma_i = \phi_i$ to get a *Shapley global impact* $\Phi_i$, or $\gamma_i = \beta_i$ to get a *Banzhaf global impact* $B_i$.

For each of the datasets and algorithms we computed and plotted the Shapley and Banzhaf global impacts. The obtained plots can be found in the supplementary material.

For `NHANES`, `BOSTON`, and `VEHICLE_INSURANCE` datasets, the obtained plots of Banzhaf/Shapley global impacts, computed using `BANZHAF` and `TREESHAP_PATH` respectively, are virtually indistinguishable. For the larger instance based on the dataset `FLIGHTS`, only very small differences in the ordering of features by importance can be observed for both `FLIGHTS_GB` and `FLIGHTS_DT`.

**Specific data points.** We now turn to describing how much the obtained Banzhaf and Shapley attributions deviate from each other for specific data points. To measure the difference between the feature orderings produced by both methods, we computed the *modified Cayley distance* between the respective orderings of $n \in \{3, 10, 20\}$ most important features for each data point, and took the average over all data points. The Cayley distance measures the number of swaps needed to switch from one permutation to another. In our modified version, we also support the case where the sets of considered most important features in the respective permutations are different. For a missing feature, we add it at the end of the permutation. The results are presented in Table 3. They confirm that the differences are on average small; in particular for the instances `BOSTON_GB`, `NHANES_GB`, and `VEHICLE_INSURANCE_GB`, for 98% of the data points, the respective 3 top features and their order matched. The orderings deviation was generally larger for DT instances where larger tree depths were allowed.

We also studied per-feature average differences between the values of Banzhaf and Shapley attributions for each

| Ins/n | 3 | 10 | 20 | Ins/n | 3 | 10 | 20 |
|---|---|---|---|---|---|---|---|
| BOS_GB | 0.02 | 1.05 | | BOS_DT | 0.08 | 1.7 | |
| NH_GB | 0.01 | 0.34 | 1.53 | NH_DT | 0.29 | 3.69 | 10.79 |
| VI_GB | 0.02 | 0.73 | | VI_DT | 0.13 | 2.60 | |
| FL_GB | 0.4 | 3.08 | 8.63 | FL_DT | 0.18 | 3.38 | 10.59 |

Table 3: The average modified Cayley distance for the $n$ most important features for $n \in \{3, 10, 20\}$ produced by `BANZHAF` and `TREESHAP_PATH` algorithms.

of the datasets. We consider both MAD (Mean Average Difference) and RMSD (Root Mean Square Difference). The relevant plots can be found in the supplementary material. Formally, for each dataset $\mathcal{D}$ out of those and each feature $i$ used therein, these are defined as: $\text{MAD}_i = \frac{1}{|\mathcal{D}|} \sum_{x \in \mathcal{D}} |\phi_i(x) - \beta_i(x)|$ and $\text{RMSD}_i = \sqrt{\frac{1}{|\mathcal{D}|} \sum_{x \in \mathcal{D}} (\phi_i(x) - \beta_i(x))^2}$. Here, $\phi_i(x)$ denotes the Shapley attribution of $f(x)$ for data point $x \in \mathcal{D}$, as computed by `TREESHAP_PATH`. Similarly, $\beta_i(x)$ denotes the Banzhaf attribution as computed by `BANZHAF`.

For the "smaller" instances `BOSTON_GB`, `NHANES_GB`, and `VEHICLE_INSURANCE_GB` and all features, the observed MAD and RMSD differences did not exceed 5% of the corresponding global impacts. For the remaining larger models, the MAD difference did not exceed 20% for the top features. On the other hand, for the large-depth `FLIGHTS_DT` model, the RMSD difference reached around 50% even for the top features, which suggests there were data points with very big absolute differences in the produced attributions. These differences indicate that when looking at specific data points one should expect only small differences in the ordering of features and only for features with similar scores. The differences are expected to be larger for larger models.

The average error statistics also show an interesting phenomenon that, for the studied datasets and models, the per-feature Banzhaf and Shapley attributions are very close to each other even though the Banzhaf value does not satisfy the *Efficiency axiom* (in contrast to the Shapley value) and thus the sum of the produced feature scores does not typically sum up to the difference between the prediction and the "baseline" mean prediction $\mathbb{E}[f(X_U)]$.

## 4.4 NUMERICAL ACCURACY

The fact that the more significant differences between the obtained importances arised for large models suggested that the compared attribution algorithms might suffer numerical problems. To investigate this possibility and compare numerical stability of `BANZHAF` and `TREESHAP_PATH`, we considered a simple artificially prepared instance `SYNTHETIC_SPARSE` for which we know the answer for both the Shapley value and the Banzhaf value.

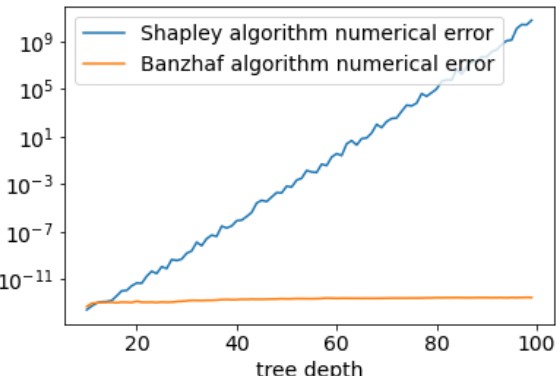

Figure 1: The numerical error for SYNTHETIC_SPARSE.

In the SYNTHETIC_SPARSE instance, the set of features is $U = \{1, \ldots, d\}$, where $d$ is a depth parameter. The instance contains one tree and one data point $x = [1, \ldots, 1] \in \mathbb{R}^d$. The tree consists of two subtrees of the same shape and depth $d - 1$. All values $f(l)$ in the leaves are equal to $0$ and $777$ in the left and the right subtree of the root, resp. All leaves $l$ have coverages equal to $33$. Every internal node of depth $i$ has one leaf child, and one non-leaf child, whose (inductively defined) subtree has depth $i - 1$. The split condition in an internal node at depth $i$ is $x_{d-i} < 1$. In this instance, the only feature with a nonzero Shapley/Banzhaf importance, equal to $388.5$, is the feature $d$ used to split at depth $0$. All other features have importances equal to $0$.[8]

We have observed that for trees of depth $d \approx 50$, errors dominate the results, i.e., the relative error exceeds $1$. In Figure 1 we visualise the mean absolute errors for TREESHAP_PATH and BANZHAF for the SYNTHETIC_SPARSE instance.

We now give a potential reason why the Banzhaf value-based implementations may be much more stable in terms of the produced relative errors. Recall that the values $\beta(l, F_l)$ for all $l \in \mathcal{L}(\mathcal{T})$, $i \in F_l$ are computed via dynamic programming using Lemmas 3 and 4. Hence, they are all computed via multiplications and divisions on *positive* numbers roughly between $0.5$ and $r_\rho$. In fact, the intermediate values $\beta(v, F_v)$ can be obtained via $O(1)$ applications of Lemmas 3 and 4 from the "parent" value $\beta(p_v, F_{p_v})$. Such a computation can be proven to introduce a multiplicative error between $1/(1 + \epsilon)^{O(1)}$ and $(1 + \epsilon)^{O(1)}$, where $\epsilon$ is the machine epsilon. This in turn implies a relative error bound of $(1 + \epsilon)^{O(1)} - 1$. Moreover, by induction on the tree depth, we can easily obtain (see the suppl. material for a proof):

**Lemma 6.** *The leaf values $\beta(l, F_l)$ can be computed with relative error at most $(1 + \epsilon)^{O(D)} - 1$.*

This bound is quite pessimistic and at the same time not very large if double precision is used and the tree depth

$D$ is small enough. On the other hand, if one considers computing the Shapley value attributions, if one wants to retain the $O(LD^2)$ time bound of the TREESHAP_PATH algorithm [Lundberg et al., 2020], then it seems that *subtractions* of intermediate values are inherent. Roughly speaking, this is because for Shapley-based attributions, if one applies an analogous dynamic programming approach, then the Shapley-analogue of Lemma 3 involves a recursive formula that is a *sum* of two "earlier" dynamic programming cells.[9] Recall, however, that our (and also Lundberg et al.'s) approach also required *inverse* applications of Lemma 3, especially when a single feature may appear multiple times on a root-leaf path. For Shapley value such an inverse application involves subtraction of equally-signed numbers.[10]

It is unclear if a similar (to Lemma 6) relative error bound can be proven in presence of such subtractions, which in general may lead to so-called *catastrophic cancellations*.

## 5 CONCLUSIONS

The contribution of this paper is twofold. First, we have developed an efficient algorithm for computing feature importance measures for tree ensemble models that is based on the Banzhaf value. This result improves the running time of previous state of the art. Second, we have presented the first extensive comparison between the Shapley and Banzhaf values in this context. We observe that both methods deliver attributions of essentially the same strength by returning almost the same ordering of features. However, these experimental results indicate that the Banzhaf value has an important advantage over the Shapley value, i.e., it allows for faster algorithms as well as these algorithms make much lower numerical errors.

We stress that this work identifies some computational/practical advantages of using the Banzhaf value compared to the Shapley value for feature attribution in tree ensemble models (in particular, the algorithm by Lundberg et al. [2020] that is commonly used by the practitioners). It would be also very interesting to compare the Shapley-based and Banzhaf-based attributions qualitatively. We believe that such a comparison requires a much more exhaustive study and is beyond the scope of this paper. However, it is, in our opinion, a very a compelling direction for future research.

**Acknowledgements**

This work has been partially supported by the ERC CoG grant TUgbOAT no 772346 and NCN project no 2020/37/B/ST6/04179.

We thank the anonymous reviewers for useful comments.

---

[8]This follows by the *sensitivity* axiom (see, e.g., [Janzing et al., 2020]) that both Banzhaf and Shapley values satisfy.

[9]See the supplementary material for details.

[10]In the original TREESHAP algorithm subtractions of this kind manifest in line 31 of [Lundberg et al., 2019, Algorithm 2].

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
