# OpenReview forum: "Improved Feature Importance Computation for Tree Models Based on the Banzhaf Value"
_auai.org/UAI/2022/Conference — UAI 2022 Poster_

### Official Review · Reviewer_oBDK · 2022-04-09

**Q2(1) Originality/Novelty:** 3
**Q2(2) Significance/Impact:** 2
**Q2(3) Correctness/Technical Quality:** 3
**Q2(6) Clarity Of Writing:** 3
**Q6 Overall Score:** 6
**Q8 Confidence In Your Score:** 4

**Q1 Summary And Contributions:**

The paper addresses the question of attributing an importance index to features for tree models. This problem can be seen as a game theoretic question usually answered by computing the Shapley values of the features, but that the Banzhaf value offers an attractive alternate path. Indeed, the paper shows how it can be computed in an elegant way that is both faster and less prone to the accrual of computational error than Shapley's, while offering so called explanations that are mostly the same.

**Q2 Assessment Of The Paper:**

More detailed information regarding each of these aspects is given below:

**Q2(4) Quality Of Experiments (Optional):**

2: Fair: The experimental evaluation is weak: important baselines are missing, or the results do not adequately support the main claims.

**Q2(5) Reproducibility:**

3: Good: Key resources (e.g., proofs, code, data) are available and key details (e.g., proofs, experimental setup) are sufficiently well-described for competent researchers to confidently reproduce the main results.

**Q3 Main Strengths:**

The main strength of the paper is the algorithmic study of the problem of computing efficiently the Banzhaf value in a tree model. It relies on a dynamic programming approach which is clearly and convincingly detailed, and which deserves publication (but maybe not in UAI).

**Q4 Main Weakness:**

The paper is slightly out of scope. It belongs clearly to the domain of algorithmic game theory - more precisely, transferable utility games where the question is about how to share the value of a coalitional game - but does not really deal with uncertainty.
The role of the Banzhaf value is not clear: it is presented as an alternative to Shapley's, but is evaluated as an approximation of it.
The use of these index is left unclear, corrupting the experimental study.

**Q5 Detailed Comments To The Authors:**

I detail my answer to Q4 above:
1. Regarding the approach, it seems to me that the authors consider the Banzhaf value as a convenient proxy for the Shapley value. This is theoretically untrue, and there are reasons to think the Shapley value is superior to Banzhaf for feature attribution (this is implicitly acknowledged by the authors in Appendix C). The vocabulary of "errors" to measure the dissimilarity/divergence between these two values is also telling: one is the groud truth, the other is an approximation of it. I think the paper should be clearer about this question (note that the very good algorithmic core of the study is completely agnostic to this issue).

2. Value attribution is not an "explanation", because it is not an argument that can be presented to an explainee. It is an indicator in a dashboard, which is fine, but different.

3. In turn, the experimental study of the paper is not convincing, because it is unclear about the two previous points - what should we measure to assess the quality of the output? The authors are trying to convince the reader that Banzhaf's is an acceptable substitute for Shapley's, but giving aggregated indices over a random dataset won't do the job, because the questions are: which decisions are taken on the basis of these indicators? how? how often do they diverge? how far? The first two questions are tied to the fact these are not explantions, and their influence in the decision-making process is unclear, but this is out of the scope of the paper. Yet, I think the last two questions would be better addressed with other metrics than the one chosen by the authors (maybe quantiles and boxplots rather than averages, and another way to measure divergence than the mean squared "error"). Besides, the figures in the appendix are really painful to read.

**Q7 Justification For Your Score:**

The algorithmic part is sound, clear and useful, so the paper is above the threshold.
It is not far above because it is slightly unfocused and out of scope.


**Q9 Complying With Reviewing Instructions:**

1: Yes.

---

### Official Review · Reviewer_QiHY · 2022-04-10

**Q2(1) Originality/Novelty:** 3
**Q2(2) Significance/Impact:** 3
**Q2(3) Correctness/Technical Quality:** 3
**Q2(6) Clarity Of Writing:** 3
**Q6 Overall Score:** 6
**Q8 Confidence In Your Score:** 3

**Q1 Summary And Contributions:**

This paper proposes leveraging the Banzhaf value as an alternative to the Shapley value, which has been used prominently in methods for deriving explanations of machine learning tools. The authors motivate the investigation of the BV in the context of explaining prediction tools based on tree ensembles, and show that it provides certain advantages while affording explanations that are very similar to those provided by techniques using the SV.

**Q2 Assessment Of The Paper:**

More detailed information regarding each of these aspects is given below:

**Q2(4) Quality Of Experiments (Optional):**

3: Good: The experimental evaluation is adequate, and the results convincingly support the main claims.

**Q2(5) Reproducibility:**

3: Good: Key resources (e.g., proofs, code, data) are available and key details (e.g., proofs, experimental setup) are sufficiently well-described for competent researchers to confidently reproduce the main results.

**Q3 Main Strengths:**

The main strengths of this work are the following:
-- Novelty
-- Well written
-- Practical application shown via empirical evaluation


**Q4 Main Weakness:**

The main weakness that I found is that presentation suffers somewhat given that a lot of material that is important to understanding some aspects of the paper are pushed to the appendix. I understand that the page limit sometimes leaves no choice, but perhaps in this case the presentation could be improved.

**Q5 Detailed Comments To The Authors:**

No further comments in addition to those made in the other sections of the review.

**Q7 Justification For Your Score:**

The paper proposes a novel method, which is evaluated empirically. The main concern is presentation, since the authors include more than twice the number of pages in the main body as appendices.

**Q9 Complying With Reviewing Instructions:**

1: Yes.

---

### Official Review · Reviewer_eNzF · 2022-04-11

**Q2(1) Originality/Novelty:** 2
**Q2(2) Significance/Impact:** 2
**Q2(3) Correctness/Technical Quality:** 3
**Q2(6) Clarity Of Writing:** 3
**Q6 Overall Score:** 6
**Q8 Confidence In Your Score:** 3

**Q1 Summary And Contributions:**

The paper proposes that using Banzhaf value for generating feature importance in tree ensemble models performs better than using classical Shapley values method. The paper shows that this is the case for the time taken to generate explanations without any change in performance. The paper proposes a theoretical analysis and the experiments done on 4 data sets show that Banzhaf values are an efficient alternative to Shapley values to generate explanations for tree ensembles.

**Q2 Assessment Of The Paper:**

More detailed information regarding each of these aspects is given below:

**Q2(4) Quality Of Experiments (Optional):**

3: Good: The experimental evaluation is adequate, and the results convincingly support the main claims.

**Q2(5) Reproducibility:**

3: Good: Key resources (e.g., proofs, code, data) are available and key details (e.g., proofs, experimental setup) are sufficiently well-described for competent researchers to confidently reproduce the main results.

**Q3 Main Strengths:**

1. The paper shows a nice result and can lead to a widespread use of Banzhaf values for explaining not just tree based models but also other machine learning models given such an efficiency gain.

2. The paper is sound and the experiments shows that the method works well in practice.


**Q4 Main Weakness:**

1. It is known that the Shapley value always distributes the payoff that can be obtained by all cooperating players while the Banzhaf value does not, thus making its calculation inefficient. The discussion on this seems to be missing the paper as this can be a good motivation.

2. How does this method compare to other tree ensemble explanation methods such a TREX and MAPLE? The main question here is that why should I use the Banzhaf value based explanation when other (better) explanation methods are available?

3. It would have been nice to see experiments on random forest along with simple DTs and GBTs.

**Q5 Detailed Comments To The Authors:**

In addition to points raised in Q4., I have a few questions/comments here I would like the authors to comment on:

1.  The Shapley value allows contrastive explanations. Does a Banzhaf value as well?

2. Does Banzhaf value method suffers from inclusion of unrealistic data instances when features are correlated like Shapley values?

3. There are several relevant papers that are missing and should be cited as discussed. I list a few of them here:
a) Structural Tractability of Shapley and Banzhaf Values in Allocation Games, Greco et al., IJCAI 2015
b) TREX: Tree-Ensemble Representer-Point Explanations, Brophy and Lowd
c) On the Trustworthiness of Tree Ensemble Explainability Methods, Yasodhara et al., CD-MAKE 2021
d) MonoForest framework for tree ensemble analysis, Kuralenok et al., NeurIPS 2019

Overall, I liked the paper but the issues mentioned in Q4 and 5 prevent me to suggest fulkl acceptance at this point. Of course I am open to change my rating during rebuttal.


**Q7 Justification For Your Score:**

I read the paper completely although did not check all claims completely. I weighed the issues in Q4 and 5 while making the decision.

**Q9 Complying With Reviewing Instructions:**

1: Yes.

---

### Official Review · Reviewer_pP1j · 2022-04-11

**Q2(1) Originality/Novelty:** 2
**Q2(2) Significance/Impact:** 2
**Q2(3) Correctness/Technical Quality:** 3
**Q2(6) Clarity Of Writing:** 1
**Q6 Overall Score:** 4
**Q8 Confidence In Your Score:** 2

**Q1 Summary And Contributions:**

This paper proposes to use a measure called the "Banzhalf"  value  to compute the importance of individual features.  An algorithm is proposed which complexity is linear in the size of the input. Experiments are presented, that compare the approach proposed to an algorithm computing the shapley value (but for the cpu time, the results are unclear for me)

**Q2 Assessment Of The Paper:**

More detailed information regarding each of these aspects is given below:

**Q2(4) Quality Of Experiments (Optional):**

3: Good: The experimental evaluation is adequate, and the results convincingly support the main claims.

**Q2(5) Reproducibility:**

2: Fair: Key resources (e.g., proofs, code, data) are unavailable but key details (e.g., proof sketches, experimental setup) are sufficiently well-described for an expert to confidently reproduce the main results.

**Q3 Main Strengths:**

The writing being poor,  the strengths of the paper are unclear for me.

The experiments suggest that   the computation of the "Banzhalf" value provides an approximation of the Shapley value and  is faster to obtain.

**Q4 Main Weakness:**

First of all, the writing is very poor -  even the introduction is not readable for a reader who is not a specialist of the domain.  Section 2 (preliminaries)  does not provide more information - we dont know what a tree ensemble is and why computing the importance of attributes is valuable (to what extend does this computation provide an "explanation"). Likewise, we dont know what should be the properties of the measure of importance searched for.

The motivation are also unclear - the fact that the shapley value is used but is not the only solution concept in game theory is not a scientific claim to defend the proposition of any other measure coming from the domain of game theory.

**Q5 Detailed Comments To The Authors:**

The presentation of the paper, and especially sections 1 and 2 have to be rewritten so that any reader can understand the problem

We also need a more detailed study of the properties of the Banzhalf  value (compared to the "attractive properties" of the Shapley values - which are ?)

If the goal is to propose the use of the Banzhal value instead of the Shapley one (as an approximation ? the paper suggests that the ordering on features is generally  to be the same one), the quality of this approximation has to be studied, both from the theoretical and the experimental point of view.

**Q7 Justification For Your Score:**

As written above, the paper is very unclear.  We main result is an algorithm for computing the Banzhalf value, but the significance of this value for the domain targeted is still to be shown.

**Q9 Complying With Reviewing Instructions:**

1: Yes.

---

### Decision · Program_Chairs · 2022-05-15

**Decision:**

Accept (Poster)

**Comment:**

Meta Review: Pros:
1. The paper contributes some new ideas.
2. The paper is likely to have a moderate impact within a subfield of AI.
3. The experimental evaluation is adequate, and the results convincingly support the main claims.
4. The paper is well organized but the presentation could be improved.
5. Empirical evaluation

Cons:
1. The exposition could be improved.
2. Need to better understand the differences between Shapley and Banzhaf values to compute feature importance values.